

# Differential Temperature Sensitivity of Intracellular and Extracellular Soil Enzyme Activities.

Adetunji Alex Adekanmbi[1,2], Laurence Dale[1], Liz Shaw[1], and Tom Sizmur[1]

[1]Department of Geography and Environmental Science, University of Reading, Reading, RG6 6DW, UK

[2]Department of Soil Science and Land Management, Federal University of Technology, PMB 65, 920001, Minna, Nigeria.

*Correspondence to*: Tom Sizmur (t.sizmur@reading.ac.uk)

**Abstract.** Predictions concerning the feedback of soil heterotrophic respiration to a warming climate often do not differentiate between the extracellular and intracellular processes involved in soil organic matter decomposition. This study examined the temperature sensitivities of intracellular and extracellular soil enzyme activities and how they are influenced by previous temperatures. We pre-incubated soils at 5 °C, 15 °C or 26 °C to acclimatise the microbial communities to different thermal regimes for 60 days before measuring potential activities of β-glucosidase and chitinase (extracellular
enzymes), glucose-induced respiration (intracellular enzymes), and basal respiration at a range of assay temperatures (5 °C, 15 °C, 26 °C, 37 °C, and 45 °C). A higher pre-incubation temperature decreased soil pH and C/N ratio and decreased β-glucosidase potential activity and respiration, but not chitinase potential activity. It is likely that this legacy effect on β-glucosidase and respiration is an indirect effect of substrate depletion rather than physiological acclimatation or genetic adaptation. Pre-incubation temperature effects on temperature sensitivity were subtle and restricted to extracellular activities,
perhaps because of the short (60 day) duration of the pre-incubation at temperatures that were below the initial optimum (~30 °C) for the mesophilic soil community. However, we found that the intracellular and extracellular enzyme activities differ in their temperature sensitivity and this observation differs depending on the range of temperature used for Q10 estimates of temperature sensitivity. Between 5 °C and 15 °C intracellular and extracellular enzyme activities show equal temperature sensitivity, but between 15 °C and 26 °C intracellular enzyme activity was more temperature sensitive than
extracellular enzyme activity and between 26 °C and 37 °C extracellular enzyme activity was more temperature sensitive than intracellular enzyme activity. This result implies that depolymerisation of higher molecular weight carbon is more sensitive to temperature changes at higher temperatures (e.g. higher temperatures on extremely warm days) but the respiration of the generated monomers is more sensitive to temperature changes at moderate temperatures (e.g. mean daily maximum soil temperature). Therefore, since climate change predictions currently indicate that there will be a greater
frequency and severity of hot summers and heatwaves, it is possible that global warming may reduce the importance of extracellular depolymerisation relative to intracellular catalytic activity as the rate limiting step of soil organic matter mineralization. We conclude that extracellular and intracellular steps are not equally sensitive to changes in soil temperature



and that the previous temperature a soil is exposed to may influence the potential activity, but not temperature sensitivity, of extracellular and intracellular enzymes.


## 1 Introduction

Understanding the temperature sensitivity of soil organic matter (SOM) decomposition will help predict how soils might respond to climate change. There are two major enzymatically-mediated steps involved in the decomposition of SOM to produce $CO_2$ (Bárta et al., 2013; Maire et al., 2013; Blagodatskaya et al., 2016). The first step, extracellular

depolymerisation, requires extracellular enzymes of microbial (and also plant and animal) origin to depolymerize macromolecular constituents of SOM and produce soluble low molecular weight microbial substrates (Maire et al., 2013). The second step, intracellular metabolism, results in the release of $CO_2$ after substrates are absorbed and catabolised by microbial cells, involving a multitude of intracellular enzymes.

Many ecological studies have examined the temperature sensitivity of SOM decomposition, but most of them measure the end product as respired $CO_2$ (e.g. Wang et al., 2013) or mass loss of C substrate (e.g. Kirwan et al., 2014), which does not differentiate between the temperature sensitivity of contributing extracellular and intracellular processes. Temperature sensitivity, defined as the rate of change in reaction rate with respect to temperature, is the first derivative of the relationship between temperature and reaction rate (Alster et al., 2020). Temperature-rate relationships are typically unimodal, reflecting

rising reaction rates with temperature due to thermodynamic effects and then a decline in rate with further increase in temperature related to thermal effects on enzyme activation and ultimately, denaturation (Alster et al., 2020). Parameters describing the temperature-rate relationship have been shown to vary both with respect to extracellular enzyme type and, between microbial taxa for the same enzyme type (Alster et al., 2016). Extracellular, rather than intracellular, enzyme activity is widely thought to be the rate-limiting step for respiration of organic matter in soils (Jan et al., 2009: Bradford,

2013), but very few studies have explicitly compared the temperature sensitivity of extracellular and intracellular enzymes to understand how each step might respond to increases in temperature and whether the magnitude of dependence of intracellular catabolism and $CO_2$ respiration on extracellular enzyme activities for supply of substrate increases or decreases with increasing temperature. Ultimately, this lack of information limits our predictive understanding of how the soil carbon cycle will respond to future global temperature changes (Blagodatskaya et al., 2016).


As already stated, decomposition of SOM is a function of the heterotrophic microbial community and the extracellular enzymes it produces. If the microbial community and its enzyme production adapts to warming (or cooling), this might result in variation in the size of microbial enzyme (and biomass) pools (Fanin et al., 2022). In addition, thermal adaptation of the microbial community may, for a given enzyme-catalysed reaction, modulate the temperature- reaction rate relationship





(e.g. manifest as a shift in the temperature optimum of reaction rates) and thus temperature sensitivity of extra- and intra-cellular processes depending on soil thermal history (Wallenstein et al., 2010). Adaptation at the level of the microbial community may be through acclimation (phenotypic or physiological change to respond to thermal regime, including production of different isozymes within taxa), evolutionary changes within taxa leading to novel isozymes, or, species sorting where taxa (including their enzyme systems) already better adapted to a certain temperature competitively exclude

those less adapted (Birgander et al., 2013). Whether adaptive processes modulate the activity and temperature response relationship to the same extent for intracellular and extracellular processes is not known. Since extracellular enzymes catalyse what is believed to be the rate limiting step in SOM decomposition (Duly and Nannipieri, 1998; Alvarez et al., 2018), any thermal adaptation of extracellular enzymes will then determine how much substrate is available for subsequent uptake and respiration and represents an important control on the response of ecosystems to warming (Bradford, 2013).


In this study, we measured potential extracellular and intracellular enzyme activities at 5 assay temperatures (5 °C, 15 °C, 26 °C, 37 °C and 45 °C) following pre-incubation for 60 days at 5 °C, 15 °C, or 26 °C. The aim was to compare the temperature sensitivity of extra- and intracellular processes related to organic matter decomposition in soils that have previously been incubated at thermal regimes, alongside measurements of key soil properties that we consider may lead to changes in

potential enzyme activity. The pre-incubation temperatures were selected to be realistic for the site where the soil was sampled. We hypothesise that (i) extracellular and intracellular enzyme activities are not equally sensitive in their response to increasing temperature, given the involvement of different enzymes and (for intracellular catabolism) biochemical networks, and that (ii) extracellular and intracellular activities and their temperature sensitivity   are influenced by pre-incubation temperature due to thermal adaptation of the soil microbial community.


## 2 Methodology

### 2.1 Soil sampling and pre-incubation

Soils samples were collected from a depth of 3 – 10 cm from a permanent grassland field at Sonning UK (latitude 51° 28.564′, longitude 000° 54.198′), sieved with a 4 mm sieve, mixed, and homogenised before randomly allocating to

replicates. Four 'field moist' (soil moisture content = 0.13 g $H_2O$ g soil$^{-1}$) replicate sub-samples (750 g) were pre-incubated at 5 °C, 15 °C (similar to the mean daily minimum (6.8 °C) and the mean daily maximum (15.0 °C) temperatures measured at the University of Reading Atmospheric Observatory, close to the sampling location, between 2009 and 2019), and 26 °C (typical of a temperature measured on a warm summer day at the University of Reading Atmospheric Observatory; Figure S-1) for a period of 60 days in plastic containers with the cover of each container loosely closed. Soil moisture content was

adjusted to the initial field moist condition every two weeks for soils incubated at 5 °C and 15 °C and weekly for soils incubated at 26 °C. The soil is a slightly acidic loamy soil, classified as Chromic Endoskeletic Luvisol. A detailed description of the site is provided by Adekanmbi et al., (2020)(Adekanmbi et al., 2020).



## 2.2 Experimental design

The experiment was a two factorial experiment involving the 3 pre-incubation temperatures (5 °C, 15 °C, and 26 °C), and 5
assay temperatures (5 °C, 15 °C, 26 °C, 37 °C and 45 °C). This design resulted in 15 treatments replicated 4 times, resulting
in 60 experimental units. At the end of the 60 day pre-incubation period, soils were subsampled for determination of basal
respiration and substrate induced respiration using glucose as the substrate (Section 2.3), and the potential activity of β-
glucosidase (β-1,4-glucosidase) and chitinase (N-acetyl β – D – glycosaminidase) extracellular enzymes (Section 2.4).
Assays were performed on all experimental units within the same week to minimise variability due to time of assay.
Incubation temperatures were randomised to prevent systematic bias in the results. A portion of the soil from each replicate
sample was also analysed for total C, total N, pH, and microbial biomass carbon (Section 2.5).

## 2.3 Basal and substrate-induced respiration

For each replicate sample, 15 g (13.31 g dry weight equivalent) of soil was weighed into a 50 ml centrifuge tube. Glucose
solution (2 ml) was added at concentrations of 0 (deionized water only) or 10 mg $g^{-1}$ soil (an initially saturating concentration
of glucose as determined in a preliminary experiment; supplementary information), thus bringing the soil to 58 % of its water
holding capacity. The soil was then mixed to distribute the solution throughout. Following soil-substrate mixing, the tube
was ventilated by blowing in lab air with a 20 ml syringe.  The tubes were then sealed with septum stoppers and 15 ml of lab
air was injected. The headspace was flushed by moving the syringe plunger up and down several times before sampling 15
ml of head space gas (as the T0 sample) and injecting into an evacuated 12 ml exetainer vial, creating overpressure, using a
tap and needle attached to the syringe. Soil samples were incubated for one hour at either 5 °C, 15 °C, 26 °C, 37 °C or 45 °C.
At the end of the incubation, the process of injecting air, flushing, and sampling was repeated (T1 sample). Headspace gas
samples were stored at 20 °C prior to analysis by an Agilent 7890B gas chromatograph. After calibrating with $CO_2$ gas
standards, the concentration of $CO_2$ in mg $L^{-1}$ was converted to C-$CO_2$ mg C $g^{-1}$ h as described by (Salazar-Villegas et al.,
2016):

$$CO_2 \ (\text{mg C g}^{-1} \ \text{h}^{-1}) = \frac{V \ (T1 - T0)}{Wt}$$

Where V = volume of headspace in the centrifuge tube; T1 is $CO_2$ concentration after a 1 hour incubation in mg $L^{-1}$; T0 is
$CO_2$ concentration before 1 hour incubation in mg $L^{-1}$, W is the dry weight of the soil, and t is the time between T0 and T1
measurements in hours.

## 2.4 Extracellular enzyme assays

Extracellular enzyme assay methods were based on Eivazi and Tabatabai, (1988) and Parham and Deng, (2000) for β-1,4-
glucosidase (β-glucosidase) and N-acetyl β – D – glycosaminidase (chitinase), respectively. For each experimental replicate,
1 g of soil was weighed into a 50 ml centrifuge tube and mixed with 4ml MUB buffer (pH 6) and either 1ml 25mM *p-*



nitrophenyl-β-D-glucopyranoside or 10 mM $p$-nitrophenyl-N-acetyl-b-D-glucosaminide solution, to assess β-glucosidase and
chitinase activity, respectively. Samples were incubated at 5 °C, 15 °C, 26 °C, 37 °C, or 45°C for 30 minutes, after which 1
ml 0.5 M $CaCl_2$ and 4 ml Tris buffer (pH 12) was added to stop the reaction. Samples were mixed by swirling, then filtered
with Whatman No. 2 filter paper. Additionally, 2 blanks (for each run) were created by adding substrate to tubes containing
the mixture after the reaction had stopped. Colour intensity of the filtrate was measured using a spectrophotometer at 400 nm
and blank-corrected sample absorbance converted to µg $p$-nitrophenol reaction$^{-1}$ using $p$-nitrophenol standard solutions (0,
10, 20, 30, 40 and 50 µg $p$-nitrophenol).  Potential enzyme activities were expressed as µg $p$-nitrophenol g$^{-1}$ dry soil h$^{-1}$. The
30-minute assay incubation time was within the time range where product accumulation was linear with incubation time,
according to a preliminary experiment (see supplementary material).

## 2.5 Measurement of total carbon, total nitrogen, pH, and microbial biomass carbon

Microbial biomass carbon was measured using the fumigation/extraction method described by Vance et al. (1987). Four
replicates from each pre-incubation temperature were weighed to the moist mass equivalent to 50 g oven-dried soil in
beakers and placed in a vacuum desiccator lined with damp paper towel to ensure high humidity, along with a beaker
containing about 50 ml ethanol-free chloroform and several anti- bumping granules. The desiccator was evacuated, and the
chloroform allowed to boil for two minutes before the valve was closed and the desiccator kept in the dark for 24 hours.
Before extraction, the chloroform was removed, the desiccator evacuated three times and the samples left to vent to ensure
no chloroform remained in the soil.

Extraction was carried out on both fumigated soil and non-fumigated duplicates. Samples of both were placed into 350 ml
polypropylene bottles, to which 200 ml 0.5 M $K_2SO_4$ was added, before being placed on an oscillating shaker for 30 minutes.
The suspension was then filtered into polypropylene universal tubes before being stored in a freezer prior to analysis. After
removal from the freezer, samples were diluted by a factor of 10, and filtered to remove $CaSO_4$ that had precipitated, before
analysis for total organic carbon (TOC) using a Shimadzu TOC 5000. Also analysed were method blanks consisting of
$K_2SO_4$ that had not been used to extract soil, to correct for any part of the reading not due to organic carbon content. TOC
extracted from fumigated and non-fumigated samples was converted to a biomass carbon value by multiplying the difference
($E_c$) by 2.64, following Vance et al., (1987). The TOC of the non-fumigated soil before conversion represents the $K_2SO_4$
extractable carbon.

Total C and N were determined using the dry combustion method. 2 mm sieved soil samples were ground for three minutes
in an agate ball mill. From the residue, 10 mg duplicates were weighed out using a five-point balance and placed in tin foil
capsules for measurement. C and N concentrations were analysed using a C/N Elemental Analyser (Thermo Flash 2000 EA).
The C/N ratio was calculated from total C and N.





pH was determined in water (10 g air-dried soil: 25ml deionised water) following end-over-end shaking (30 rpm, 15 mins) and using a calibrated (pH 4.0 and pH 7.0) pH meter.

### 2.6 Temperature sensitivity

Temperature sensitivity (Q10) of both the intra (glucose induced respiration) and extra-cellular (chitinase and β-glucosidase) enzyme activities was calculated using the equal time measurement method, as described by Karhu et al., (2014). Q10 was calculated at three temperature ranges ($Q10_{5-15°C}$, $Q10_{15-26°C}$, and $Q10_{26-37°C}$). Arrhenius enzyme activation energy (Ea) was calculated from the slope of the relationship between $-1/R_0T$ and the natural logarithm of rate of enzyme activity ($R_0$ = the gas universal constant: 8.314 J mol$^{-1}$; T = temperature in Kelvin), as described by Li et al., (2015). Ea was calculated for two ranges of temperature (5 °C – 26 °C for intracellular enzymes and basal respiration, and 5 °C – 37 °C for the two extracellular enzymes) to ensure that the data used to calculate Ea conformed to the Arrhenius functional form (Schulte, 2015).

### 2.7 Statistical analysis

Two-way Analysis of Variance (ANOVA) was carried out to assess the effects of pre-incubation temperature and assay temperature on basal respiration, intracellular, and extracellular enzyme activities. We also assessed whether intracellular and extracellular enzymes were equally sensitive to temperature, and whether this was influenced by pre-incubation temperature, by performing a two-way ANOVA on the Ea and Q10 values using enzyme type, and pre-incubation temperature as factors. One way ANOVA was carried out to assess the effect of pre-incubation temperature on soil properties. ANOVA was performed in Minitab version 18. Tukey pairwise comparisons were used to assess the significance of differences between individual treatment means.

### 3 Results

### 3.1 Impact of pre-incubation temperature on selected soil properties

The effects of soil pre-incubation temperature on soil total C, total N, C/N ratio, pH and microbial biomass carbon are presented in Figure 1. Pre-incubation temperature did not have a statistically significant impact on C ($P = 0.641$) or N ($P = 0.439$). However, soil C/N ratio was significantly ($P < 0.05$) higher in soil pre-incubated at 15 °C and 5 °C, compared to soil pre-incubated at 26 °C. Also, pre-incubation temperature significantly ($P < 0.05$) influenced soil pH which decreased in the order 5 °C > 15 °C > 26 °C. There was no statistically significant effect of soil pre-incubation temperature on soil microbial biomass ($P = 0.206$).



## 3.2 Responses of intracellular and extracellular enzyme activities to pre-incubation temperature and assay temperature.

The influence of pre-incubation temperature on the potential activities of β-glucosidase (Figure 2A) and chitinase (Figure 2B) extracellular enzymes, the rate of glucose-induced respiration (representing the potential intracellular enzyme activity) (Figure 2C), and the basal respiration rate (Figure 2D) across the full range of assay temperatures (5 °C to 45 °C) are presented in Figure 2.

Both the pre-incubation temperature ($P < 0.0001$), assay temperature ($P < 0.0001$) and their interaction ($P = 0.001$) significantly influenced potential β-glucosidase activity in soil. Soils pre-incubated at 26 °C had a lower potential β-glucosidase activity in soil compared to those pre-incubated at 15 °C or 5 °C. Increasing assay temperature increased β-glucosidase activity up to the maximum assay temperature of 45 °C (Figure 2A). Pre-incubating soils at 15 °C resulted in significantly greater potential β-glucosidase activity at the higher assay temperatures (45 °C and 37 °C) than by pre-incubating soils at 5 °C or 26 °C.

Both assay temperature ($P < 0.0001$) and interaction between assay and pre-incubation temperatures ($P < 0.001$) significantly influenced potential chitinase activity, but pre-incubation temperature ($P = 0.077$) did not. Chitinase activity increased with increasing assay temperature, reaching maximum when assayed at 37 °C, but was then lower when assayed at 45 °C (Figure 2B). Pre-incubating soil at 26 °C and assaying at 37 °C resulted in a significantly ($P = 0.001$) greater chitinase activity than pre-incubating at 5 °C assaying at 37 °C. When assayed at 5 °C or 15 °C, pre-incubation at 26 °C resulted in lower chitinase activities than pre-incubation at 15 °C or 5 °C.

Both the pre-incubation temperature ($P < 0.0001$), and assay temperature ($P < 0.0001$) significantly influenced glucose-induced respiration, but not their interaction ($P = 0.130$). Similarly, the pre-incubation temperature ($P = 0.001$), and assay temperature ($P < 0.0001$) significantly influenced basal respiration, but not their interactions ($P = 0.250$). With or without glucose addition, pre-incubating soil at 26 °C resulted in lower soil respiration compared to pre-incubating soil at 5 °C or 15 °C (Figure 2C and 2D). Glucose-induced respiration increased with increasing assay temperature, reaching maximum between 26 °C and 37 °C, but was significantly lower at 45 °C. Basal respiration increased with increasing assay temperature up to 26 °C then declined only slightly. The addition of 10 mg g$^{-1}$ soil of glucose led to about a 4-fold increase in $CO_2$ respired, compared to no addition of glucose substrate.

## 3.3 Effect of pre-incubation temperature and enzyme type (intra- or extracellular) on temperature sensitivity of potential enzyme activity





### 3.3.1 Temperature coefficient (Q10)

The effects of pre-incubation temperature and enzyme type on $Q10_{5\text{-}15°C}$, $Q10_{15\text{-}26°C}$, and $Q10_{26\text{-}37°C}$, are presented in Figure 3. There was no overall significant effect ($P > 0.05$) of pre-incubation temperature on Q10 calculated using all three temperature intervals. There was also no significant effect of enzyme type ($P = 0.393$), or the interaction between enzyme type and pre-incubation temperature ($P = 0.700$), on $Q10_{5\text{-}15°C}$. However, $Q10_{15\text{-}26°C}$ significantly differed with enzyme type ($P < 0.0001$), but not for the enzyme type*pre-incubation interaction ($P = 0.160$). The $Q10_{15\text{-}26°C}$ was significantly lower for

both extracellular enzymes (chitinase and β-glucosidase) than for intracellular enzyme activity (glucose-induced respiration) or basal respiration, irrespective of pre-incubation temperature. This result indicates that intracellular enzymes are more temperature sensitive than extracellular enzymes in this soil between 15 °C and 26 °C. Furthermore, $Q10_{26\text{-}37°C}$ was significantly affected by enzyme type ($P < 0.0001$) but exhibited the opposite pattern to $Q10_{15\text{-}26°C}$. $Q10_{26\text{-}37°C}$ for chitinase activity and β-glucosidase activity were significantly ($P < 0.05$) greater than the $Q10_{26\text{-}37°C}$ for intracellular enzyme activity

(glucose-induced respiration) and basal respiration. This finding indicates that extracellular enzymes are more temperature sensitive than intracellular enzymes in this soil between 26 °C and 37 °C. $Q10_{26\text{-}37°C}$ for chitinase activity was also significantly ($P < 0.05$) greater than the $Q10_{26\text{-}37°C}$ for β-glucosidase activity. There was also a significant interaction between enzyme type and pre-incubation temperature ($P = 0.018$). Chitinase activity was less temperature sensitive when soil was pre-incubated at 26 °C compared to when pre-incubated at 15 °C or 5 °C.


### 3.3.2 Arrhenius activation energy (Ea)

The activation energy (Ea), derived from the fit of the Arrhenius equation (Figure 4) to assays performed between 5 °C and 26 °C (basal respiration and intracellular enzyme activity) and between 5°C and 37°C (extracellular enzymes), differed significantly with enzyme type ($P < 0.0001$) and pre-incubation temperature ($P = 0.001$) and there was a significant

interaction between enzyme type and pre-incubation temperature ($P = 0.046$). Ea increased with increasing pre-incubation temperature, with soils pre-incubated at 26 °C exhibiting the highest Ea and soils pre-incubated at 5 °C exhibiting the lowest. β-glucosidase activity enzyme had a significantly lower Ea than chitinase activity, intracellular enzyme activity, and basal respiration.

### 4 Discussion

Understanding whether soil intracellular and extracellular enzyme activities, which each play a distinct role in SOM decomposition processes, are equally sensitive to temperature changes was the major motivation for this study. We also examined whether pre-incubation temperature drives thermal adaption of the soil microbial community and results in differential alteration of temperature sensitivity of intracellular and extracellular enzyme activity. Therefore, we pre-incubated soil samples at three different temperatures to expose the soil microbial community to a particular thermal regime





and then assayed intracellular (as respiration induced by a saturating concentration of glucose) and potential extracellular
enzyme activity. Because extracellular enzymes were assayed in soil slurries and also in the presence of excess substrate, it
was assumed that substrate diffusion or substrate concentration did not limit reaction rates and that the observed potential
reaction rate was thus a function of enzyme properties and enzyme concentration (Wallenstein and Weintraub, 2008).
Alongside intracellular and extracellular enzyme activity we measured basal respiration as a reference. We assume that the

rate of basal respiration will represent the intracellular activity as supplied by substrate from extracellular enzyme activity,
and that the rate of respiration may be limited by substrate availability (to extracellular processes) and its supply (to
intracellular processes) by extracellular activity and diffusion. Thus, any differential effects of pre-incubation temperature
on temperature sensitivity of basal respiration cannot be interpreted solely as a function of differences in the cellular
physiology of the microbial communities present.


Examining the general shape of the response of potential activity to assay temperature, we found that activity of β-
glucosidase increased with increasing incubation temperature to our highest assay temperature of 45 °C. Our result is
consistent with the increase in β-glucosidase activity with temperature reported in other studies using assay temperatures as
low as 2 °C and as high as 65 °C (Steinweg et al., 2013) or 70 °C (Trasar-Cepeda et al., 2007) and showing increases in

activity up to and beyond 45 °C. The potential activity of chitinase also increased with temperature, but, in contrast to β-
glucosidase, the response, over the range of assay temperatures, was unimodal, reaching a maximum activity between 37 °C
and 45 °C. This observed unimodal response to increasing temperature is interpreted in terms of three distinct phases: (i) a
rising phase where temperature increases lead to increasing reaction rate due to thermodynamic effects, (ii) a plateau which
represents the optimum temperature and (iii) a steep falling phase where rate declines beyond the optimum temperature

(Schulte, 2015) attributed to thermal denaturation of proteins. Our optimum for chitinase (37 to 45 °C) is relatively
consistent with the report of a maximum activity for soil chitinase of 45.5 °C (as assayed through quantification of N-acetyl
glucosamine released from added chitin; (Rodriguez-Kabana, et al., 1983)) but contrasts to the optimum of ~63 °C reported
in the study by (Parham and Deng, 2000) using the same p-nitrophenol-based assay as used here. Differences in these
optimum temperature-activity responses between soils may be due to differences in microbial composition (and thus

microbial-produced chitinase isozymes) between soils. Optimum temperatures varying between 40 °C and 60 °C have been
recorded for chitinases (partially) purified from soil microorganisms (Gao et al., 2008;Alster et al., 2016; Du et al.,
2021;Thakur et al., 2021). Additionally, soil-type dependent stabilization of enzyme structure against thermal denaturation
through interaction with soil surfaces might also mediate differential temperature responses (Sarkar et al., 1989). It is
presumed that β-glucosidase activity in our study soil had a temperature optimum beyond the maximum tested and our

finding that the optimum temperature for chitinase activity was lower than that of β-glucosidase is likely due to between-
enzyme family differences in protein structural properties conferring thermal stability, resulting in differential susceptibility
of different enzyme families to thermal denaturation or degree of stabilization in soil. Our finding that intracellular
catabolism increased with increasing assay temperature up to an optimal temperature between 26 °C and 37 °C, followed by



a significant decline thereafter, is very likely due to the inability of the microbial population to function optimally above 37 °C due to impairments in their physiological processes (Todd-Brown et al., 2012; Maire et al., 2013). The optimum temperature recorded here is greater than the annual average temperature but is within the range of the maximum soil temperature experienced for this soil. These findings are in agreement with other studies on temperate soils recording an optimum temperature for microbial growth of ~30 °C (Bárcenas-Moreno et al., 2009), although basal respiration rate has been shown to increase with increasing temperature to 45 °C and not to be coupled to microbial growth (Pietikäinen et al.,
295 2005).

We found that the intracellular and extracellular enzyme activities differ in their temperature sensitivity and this observation differs depending on the range of temperature used for Q10 estimates of temperature sensitivity. Intracellular catalytic activity was more sensitive to temperature changes at a moderate temperature range (15 °C and 26 °C) than extracellular
enzymes. Conversely, extracellular enzymes were more sensitive than intracellular enzymes to temperature changes at a higher temperature range (26 °C and 37 °C). These results imply that, in the soil we studied, extracellular depolymerase activity was more temperature sensitive at higher temperatures and intracellular catalytic enzyme activity was more temperature sensitive at moderate temperatures. At the site where the soil was collected for this experiment the annual mean daily maximum soil temperature was approximately 15 °C, whereas 26 °C reflected a typical hot summer day. Therefore,
assuming the absence of any thermal adaptation, we might expect intracellular enzyme potential to be more sensitive to global warming-induced increases in the mean daily maximum soil temperature, but extracellular enzymes might be more sensitive to increased maximum temperatures on extremely warm days. The findings described above support our first hypothesis that the potential rate of extracellular depolymerisation and intracellular catabolism are not equally temperature sensitive steps in the mineralisation of organic matter in soils. As far as we are aware, only one other study (Blagodatskaya
et al., 2016) has considered potential intra- and extracellular activities involved in organic matter decomposition and their responses to temperature separately. Our finding that intracellular catalytic activity is more temperature sensitive at moderate temperatures is in agreement with (Blagodatskaya et al., 2016) who calculated a $Q10_{10-20}$ for intracellular glucose oxidation of 5.1 and $Q10_{10-20}$ for chitinase and β-glucosidase activity of 1.9 and 2, respectively. Other previous research (Trasar-Cepeda et al., 2007) has compared intracellular (via dehydrogenase assay) and extracellular activity responses to a
wider range of temperatures (5-70 °C), but, not necessarily under potential (substrate-excess) conditions for intracellular activity measurement, as we have done here. Thus, further experiments are required to evaluate the applicability of our finding of a greater temperature sensitivity of extracellular activities at higher (26 °C and 37 °C) temperature ranges to other soil types.

Climate change predictions currently indicate that there will be a greater frequency and severity of hot summers and heatwaves in Europe (Meehl and Tebaldi, 2004; Christidis et al., 2015), including Southeast England, where the soil was collected for this study. Therefore, our findings imply that, in the absence of substrate availability (or other, e.g. moisture)





limitations to activity, the rate of extracellular depolymerase-catalysed reactions will increase during heatwaves to a greater extent than the rate of intracellular catalytic reactions. Depending on the relative sizes of the intra- and extracellular enzyme
pools and substrate availability, it is possible that global warming may reduce the importance of extracellular depolymerisation relative to intracellular catalytic activity as the rate limiting step of SOM mineralization under in situ conditions. Such a switch in rate limitation, if applicable generally across all extra- and intracellular reactions, would result in an accumulation of monomers and thus potential for greater losses of C from the soil profile as dissolved organic carbon, an often overlooked component of terrestrial carbon budgets (Evans et al., 2014; Cook et al., 2018).


The temperature sensitivity of C mineralisation is generally found to decrease with temperature (Niklińska and Klimek, 2007; Wang et al., 2013) and this trend has been observed in a synthesis of soil respiration measurements from laboratory studies which revealed that Q10 correlates negatively with the range of temperatures used to generate the Q10 value below 25 °C (Hamdi et al., 2013). This is consistent with kinetic theory of temperature dependence of reaction rates that explains
that the fraction of molecules with sufficient energy to react decreases in relative terms as temperature increases (Davidson and Janssens, 2006). However, similar results to our study have been reported for mineralization of (labile) C where calculated Q10 values were lower in the 0-10 or 5-15 °C range than 10-20 or 15-25 °C range, respectively (Howard and Howard, 1993; Wang et al., 2013). These findings possibly reflect that $CO_2$ production is not a function of a single non-enzyme catalysed chemical reaction but is subject to moderation by the temperature sensitivity of other components in the
involved biochemical network, for example, reduced membrane fluidity at lower temperature with implications for substrate uptake and function of membrane-embedded proteins (Schulte, 2015). Also, based on kinetic theory, it is suggested that substrates that are more recalcitrant should have higher temperature sensitivities (Davidson and Janssens, 2006). It is tempting to initially suppose that the substrates that are hydrolysed by chitinase and β-glucosidase enzymes in depolymerization reactions might be more recalcitrant than glucose and other lower molecular weight substrates for
intracellular respiration and, in consequence. the extracellular-catalysed reactions should have higher temperature sensitivities. This supposition is supported by the $Q10_{26-37°C}$ data but not for Q10 calculated using the other temperature ranges. However, it should be recognised that chitinase and β-glucosidase have relatively simple di- or tri-meric substrates in nature and are assayed using artificial and simple substrates that may not be more recalcitrant than those used in intracellular metabolism. In addition, the theoretical predictions refer to chemical decomposition reactions and not
necessarily those involving enzyme catalysis (Blagodatskaya et al., 2016)) Indeed, comparison of intracellular versus extracellular estimated activation energies (Figure 4) suggested that the extracellular enzyme substrates had similar or lower (for β-glucosidase) recalcitrance. The activation energy values we obtained were in broad correspondence with those reported in other studies (Trasar-Cepeda et al., 2007).

In addition to the differences between the temperature sensitivity of extra- and intra-cellular processes, the extracellular activities were not equally temperature sensitive to each other according to $Q10_{26-37}$ and activation energy (integrating the



temperature response between 5 and 37 °C), with chitinase being more sensitive than β-glucosidase. Previous studies have also shown that the temperature sensitivity of particular classes of enzyme differs within the same soil environment (Wallenstein et al., 2010), although specific comparisons between β-glucosidase and chitinase have not always revealed
significant differences between these enzyme classes (e.g. Nottingham et al., 2016 ; Min et al., 2014; Min et al., 2019; Wei et al., 2021) and therefore the sign and magnitude of within-soil differences may not be consistent across soil types. In the case of differential temperature sensitivity with respect to enzyme type, there are implications for temperature-dependent variation in the quality of monomeric SOM constituents supplying respiration (Wallenstein et al. 2011). In the chitinase vs β-glucosidase example here, the relative activity of these enzymes would change with temperature (assuming no change in
enzyme or substrate concentration) altering the relative production of glucose and N-acetyl-glucosamine monomers and thus C and N resource availability to soil microbial communities (Min et al., 2014).

In respect of the second hypothesis, the observation that pre-incubation at 26 °C resulted in significantly lower activity, when considered across all assay temperatures, for β-glucosidase and intracellular catalytic enzymes (as well as basal respiration),
compared to pre-incubation at 5 °C or 15 °C, suggests possible adaptation of these processes to the direct or indirect effects of temperature. The indirect effects could be due to temperature-induced changes in soil properties during pre-incubation, with consequences for soil microbial activities (Sinsabaugh, 1994; Sinsabaugh et al., 1991; Adeli et al., 2005; Sinsabaugh et al., 2008; Puissant et al., 2019). It was evident in our results that pre-incubating soils at 26 °C reduced the C/N ratio when compared to pre-incubation at 5 °C or 15 °C. This probably reflects enhanced decomposition of organic matter at the
warmer pre-incubation temperature and the resulting mass loss of $CO_2$-C and enrichment of N (on a mass basis) leading to the statistically significant effect when expressed in C/N ratio from. Temperature-induced changes in C/N ratio have been reported previously (Bárta et al., 2013;Souza and Billings, 2022). The lower intracellular enzyme activity after two months exposure to a higher pre-incubation temperature is likely due to a lower (indicated, but not statistically significant) microbial biomass (and thus a reduced intracellular enzyme pool) responding to depleted relative C availability. The lower activity of
β-glucosidase for 26 °C -pre-incubated soil most likely also reflects a lower enzyme pool size, given the nature of the potential assay used to measure reaction rate and its relationship to enzyme concentration (Wallenstein and Weintraub, 2008). It is likely that such indirect effects of pre-incubation temperature on microbial community composition enzyme pool size masks any direct thermal acclimatation or genetic adaptation of the soil microbial community. It is often found that substrate depletion plays a greater role in the response of soil microbial communities to warming than physiological or
genetic shifts (Domeignoz-Horta et al., 2022). Compared to β-glucosidase, there was less evidence of an effect of pre-incubation temperature on the concentration of chitinase (no significance of pre-incubation as a main effect). The concentration of an enzyme in soil is a function of production versus turnover rate. Accordingly, the balance between these two processes, for β-glucosidase, must have been differentially influenced by pre-incubation temperature, probably both directly and indirectly via, for example, reduced microbial biomass, and complex enzyme regulation in response to altered C
availability relative to other nutrients (Allison and Vitousek, 2005; Ferraz de Almeida et al., 2015).




Whilst pre-incubation at 26 °C reduced intracellular activity, probably linked to reductions in biomass and relative C availability, it did not lead to an alteration of community intracellular temperature response traits (i.e. the shape of the temperature response) as evidenced by the non-significant interaction between pre-incubation and assay temperature or a
pre-incubation effect on temperature sensitivity as evaluated by calculation of Q10s (Figure 3) or Ea (Figure 4). This result agrees with another study that showed minimal adaptation of the temperature response (of microbial growth) to pre-incubation temperature when the temperature was below the initial optimum (~30 °C) for the mesophilic soil community (Bárcenas-Moreno et al., 2009), although pre-incubation above the optimum led to corresponding increases in the optimum for microbial growth. Minimal adaptive response to pre-incubation substantially below the initial optimum (i.e. 5 and 15 °C
in our study) is explained in terms of a rate of species sorting (ultimately favouring a community better adapted to the pre-incubation conditions) being too slow to manifest within the 60-day pre-incubation period due to slow microbial generation times at colder temperatures (Bárcenas-Moreno et al., 2009). In contrast to intracellular activity, there was some evidence (significant pre-incubation × assay temperature interaction) of a pre-incubation effect on the temperature response for potential extracellular enzyme activity, although effects were quite subtle and only systematic with pre-incubation
temperature for chitinase where activity assayed at 37 °C decreased in order of decreasing pre-incubation temperature (also discernible in effects of pre-incubation on Q10$_{26-37°C}$ and Ea). The lower Ea for chitinase for soil samples pre-incubated at the lower temperatures is consistent with the concept that cold adaptation of microorganisms leads to the production of cold adapted enzymes, by adjustment of chemical structure of the active site, with lower activation energies (Wallenstein et al., 2011). However, the few previous experimental studies examining the temperature response/sensitivity of extracellular
enzyme potentials in soils exposed to differing thermal regimes have suggested no difference in temperature sensitivity (Schindlbacher et al., 2015; Jing et al., 2019) and therefore an absence of thermal adaptation of temperature sensitivity. However, these experiments involved long-term field-based warming treatments and it is suggested that the effects of the experimental warming were negligible against the effects of wide seasonal temperature variations (Jing et al., 2019). Other studies, however, have demonstrated seasonal changes in temperature sensitivity of extracellular enzymes (Wallenstein et al.
2011; Wallenstein et al. 2009). These changes likely result from temporal changes in production of isoenzymes (by different organisms or within the same organism transcribing alternative enzyme-encoding genes) but whether these patterns represent an adaptation to seasonally-varying temperature, or are driven by other factors that change seasonally (e.g. substrate availability) is not clear (Wallenstein et al., 2011).

**5 Conclusion**

Our results advance understanding of how SOM decomposition will change under future global warming conditions. We show that the potential rates of the intracellular and extracellular steps of SOM decomposition are not equally sensitive to changes in temperature and that individual extracellular enzymes have different temperature sensitivities. Specifically, for

the grassland soil under study, we have demonstrated that potential activities of extracellular depolymerase enzymes (β-glucosidase and chitinase) have greater sensitivity to increases in temperature in the range of temperatures experienced on

extremely warm days (between 26 °C and 37 °C) than the potential activity of intracellular enzymes involved in catabolism of monomeric (e.g. glucose) substrates to $CO_2$. Since a greater prevalence of extremely hot days and heatwaves are predicted, the importance of intracellular activity may increase and the importance of extracellular activity may decrease as the rate limiting step in SOM decomposition.

For the extracellular activities studied, we found differential temperature sensitivity with respect to enzyme type. Here, the implications are for temperature-dependent variation in the quality of monomeric SOM substrates supplying respiration and potential feedbacks to soil microbial community composition given taxa-specific competitive utilization of substrates (Wallenstein et al. 2011). Whilst interpretation should be within the context of the pre-incubation conditions (60 days at temperatures less than the optimum for activity of the mesophilic community), we have also shown that the thermal history

(i.e. pre-incubation temperature) of a soil might modulate the relative responses in reaction rates to current temperature. This is both through enzyme-dependent reduction of potential activity across assay temperatures in 26 °C pre-incubated soil (for intracellular enzymes and β-glucosidase but not chitinase) and also subtle adaptation of the temperature response trait to pre-incubation temperature (for extracellular but not intracellular enzymes). Measurements of $CO_2$ alone as a response variable while studying the effect of warming may obscure our understanding of the temperature sensitivity of the

intracellular and extracellular steps of SOM decomposition.

**Data Availability**

Data associated with this publication has been uploaded to the Mendeley Data repository and is embargoed until 13[th] October

445 2023:

Adekanmbi, Adetunji Alex; Dale, Laurence; Shaw, Liz; Sizmur, Tom (2022), "Data for: Differential Temperature Sensitivity of Intracellular and Extracellular Soil Enzyme Activities", Mendeley Data, V1, doi: 10.17632/xvr3dzvdcw.1

**Author Contributions**

AAA, LD, LS, and TS conceptualised the research and designed the experiment. AAA and LD carried out the laboratory work. AAA and LD analysed the data, with support from TS and LS. AAA and LD undertook data visualisation. AAA wrote the original draft of the paper. TS, LS, LD and AAA undertook subsequent reviewing and editing. TS supervised the project and LS co-supervised the project.

**Competing Interests**

The authors declare that they have no conflict of interest



**Acknowledgements**

The authors gratefully acknowledge Technical Services staff within the Environmental Science Research Division at the

University of Reading for technical support & assistance in this work



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





**Figure 1: Effects of pre-incubation temperature on soil total carbon, total nitrogen, carbon-to-nitrogen (C/N) ratio, pH and microbial biomass carbon. Each bar and error bar represents mean and standard error of 4 replicate samples at each pre-incubation temperature. Means with the same letter are not significantly different ($P > 0.05$).**




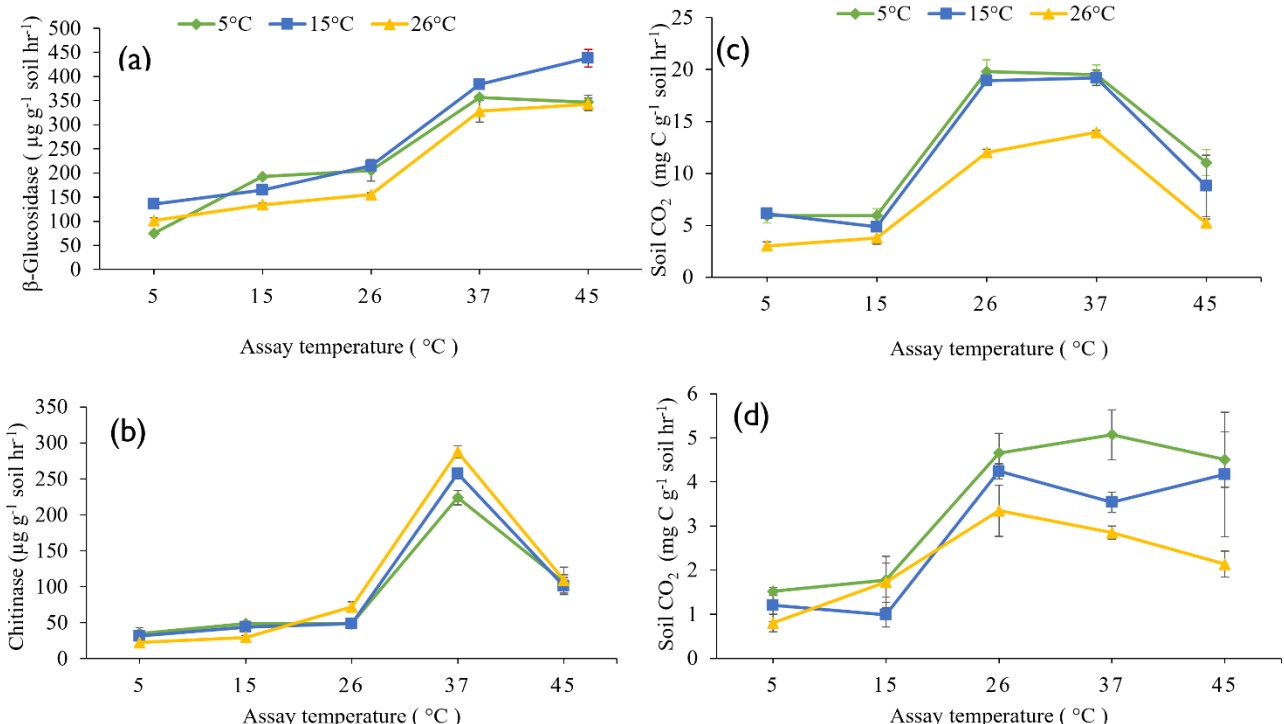

**Figure 2: Response of β-glucosidase activity (a)**, chitinase **activity (b), glucose-induced respiration (c), and basal respiration (d) to five assay temperatures (5 °C, 15 °C, 26 °C, 37 °C and 45 °C) undertaken on soils pre-incubated at three different temperatures (5 °C, 15 °C, and 26 °C). Each symbol and error bar represent mean and standard error of 4 replicate samples.**



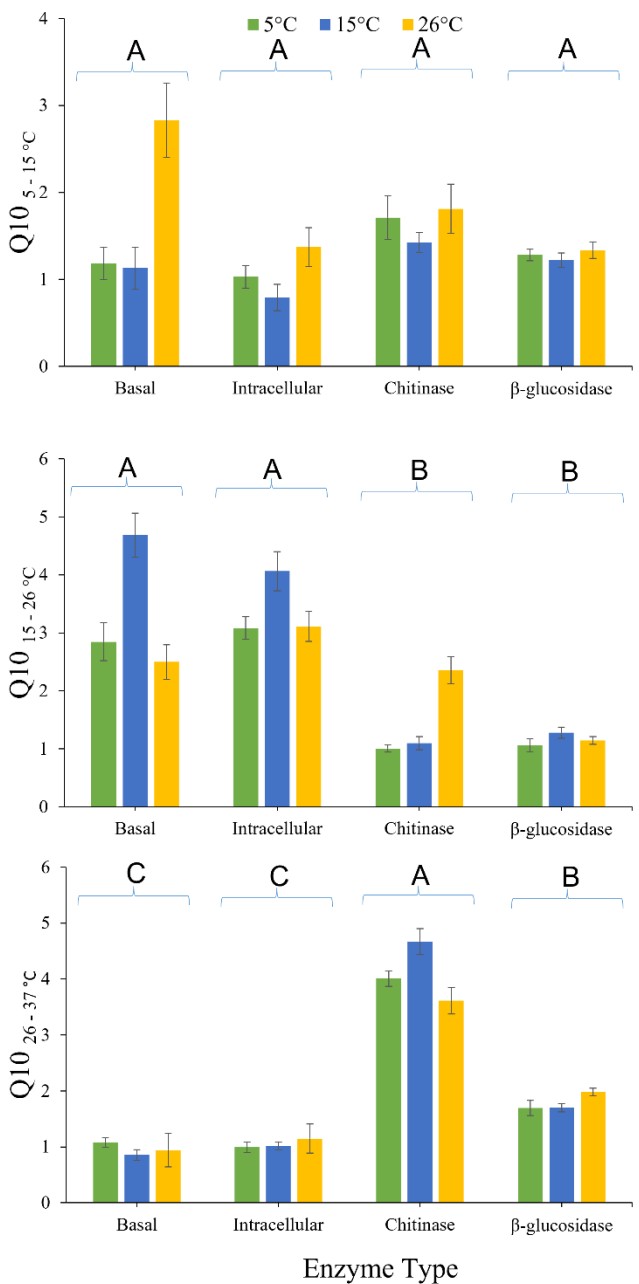

**Figure 3: Effects of pre-incubation temperature on temperature sensitivity (Q10$_{5-15°C}$, Q10$_{15-26°C}$, and Q10$_{26-37°C}$) of basal respiration rate, intracellular enzyme activity (glucose substrate induced respiration), and extracellular (chitinase and β-glucosidase) enzyme activity. Each bar and error bar represent mean and standard error of 4 replicate samples each pre-incubated at one of three different pre-incubation temperatures (5 °C, 15 °C, or 26 °C). Enzyme types sharing the same upper case letters are not significantly different ($P > 0.05$).**




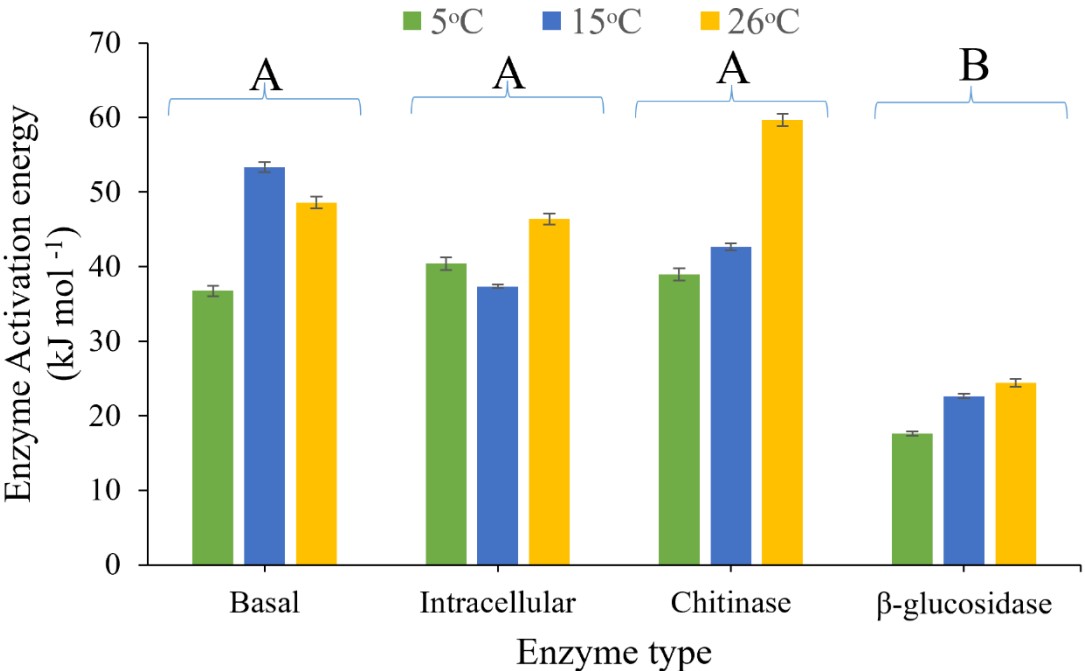

**Figure 4: Effects of pre-incubation temperature (5 °C, 15 °C, or 26 °C) on Arrhenius enzyme activation energy (Ea) for basal respiration rate, intracellular (glucose substrate induced respiration) enzyme activity, and extracellular (chitinase and β-glucosidase) enzyme activity. Ea was calculated for two ranges of temperature (5 °C – 26 °C for intracellular enzymes and basal respiration, and 5 °C – 37 °C for the two extracellular enzymes). Each bar and error bar represent mean and standard error of 4 replicates. Enzyme types sharing the same upper case letters are not significantly different ($P > 0.05$).**