# Peer review of "Differential Temperature Sensitivity of Intracellular Metabolic Processes and Extracellular Soil Enzyme Activities."

_EGUsphere, 2022_

## Author Comment (AC1)

Reviewer 1

**The article of Adekanmbi and coauthors is aiming to set up an interesting comparison between extracellular and intracellular enzymes and evaluate their temperature sensitivity after being exposed to distinct temperatures for 60 days. However, the authors need to better justify the use of glucose-induced respiration as a proxy for intracellular enzymes. Because the glucose-induced respiration will be the result of various processes and also ultimately depends on the microbial community growth efficiency. While the beta-glucosidase and chitinase activities are capturing only the activity of these enzymes. So, making the comparison between extracellular and "intracellular" enzymes becomes difficult in my understanding. Moreover, it is important to remember that the production of extracellular enzymes will also result in CO2 production. I am concerned that authors' experimental design might not allow to separate between intracellular and extracellular enzymes. Instead of referring to intracellular enzymes authors could refer to "intracellular activity" or "intracellular processes" related to SOM decomposition. This should help to avoid confusion. If authors think that the design allow to make the comparison between extracellular and "intracellular" enzymes they should add an explanation and references to justify their choice.**

**Nevertheless, I think that the data collected by the authors is valuable and is a good contribution to the field of soil ecology and to the EGU community. It could be interesting to evaluate if the respiration temperature sensitivity and extracellular enzyme temperature sensitivity are coupled or not (are they correlated?). It is also interesting to observe how distinct the two extracellular enzymes responded to the increase in temperatures. I think the authors did a good job in their discussion section.**

**It is not very clear why authors used a distinct range of temperatures to evaluate the enzyme activation energy for the respiration and extracellular enzymes. Authors could clarify this choice.**

**Overall, the paper is very well written and is citing the relevant literature in this topic.**

We thank Reviewer 1 for their positive and constructive comments on the manuscript which we address below.

**Main Comments requiring response:**

1. **The authors need to better justify the use of glucose-induced respiration as a proxy for intracellular enzymes. Instead of referring to intracellular enzymes authors could refer to "intracellular activity" or "intracellular processes" related to SOM decomposition.**

We agree with this comment (also made by reviewer 2) about the over-simplification resulting from us referring to 'intracellular enzyme activity' and propose that we revise the manuscript (including the title) so that we refer to 'intracellular metabolic processes' when speaking generally and to 'glucose-induced respiration' when speaking specifically about our results.

2. **It could be interesting to evaluate if the respiration temperature sensitivity and extracellular enzyme temperature sensitivity are coupled or not (are they correlated?).**

Thank you for this suggestion which we have explored using the Q10 data determined over the different temperature ranges. The analysis, however, revealed no relationships between either basal respiration or glucose-induced respiration and the potential extracellular activities at any of the Q10 ranges tested (please see Table). We assume the main interest here is to examine whether the temperature sensitivity of basal respiration is a function of extracellular enzyme temperature sensitivity, given that extracellular activity is usually expected to be rate-limiting to the respiration of monomers. This lack of relationship is likely due to the nature of the extracellular enzyme assays which determine potential activity (rather than in situ activity) and therefore likely do not estimate the actual rate of production of monomeric substrates for subsequent respiration. Because of the reliance on potential activity and that the correlation analysis did not reveal any possible coupling between intracellular and extracellular processes, we would prefer not to include this evaluation in revisions to our manuscript.

Table 1.  Pearson correlation coefficients for (a) $Q10_{5-10}$ (b) $Q10_{15-26}$ and (c) $Q10_{26-37}$ between basal respiration (BR), Glucose-induced respiration (GIR), potential beta-glucosidase activity and potential chitinase activity.  * denotes significant (p<0.05) correlations (n=12).

| (a) $Q10_{5-10}$ | BR | GIR | B-Gluc | Chitinase |
|---|---|---|---|---|
| BR |  |  |  |  |
| GIR | 0.758* |  |  |  |
| B-Glucosidase | -0.135 | -0.310 |  |  |
| Chitinase | -0.135 | -0.043 | 0.609* |  |
| (b) $Q10_{15-26}$ |  |  |  |  |
| BR |  |  |  |  |
| GIR | -0.143 |  |  |  |
| B-Gluc | 0.180 | 0.013 |  |  |
| Chitinase | -0.084 | 0.118 | -0.133 |  |
| (c) $Q10_{26-37}$ |  |  |  |  |
| BR |  |  |  |  |
| GIR | 0.190 |  |  |  |
| B-Gluc | 0.068 | 0.333 |  |  |
| Chitinase | 0.034 | -0.095 | -0.637* |  |

**3.  It is not very clear why authors used a distinct range of temperatures to evaluate the enzyme activation energy for the respiration and extracellular enzymes. Authors could clarify this choice.**

We found that the relationship between assay temperature and reaction rate was non monotonic. However, Arrhenius assumes that reaction rates rise monotonically with temperature and thus does not take in to account the typical unimodal response (discussed in the second paragraph in the discussion) due to declines in activity and thermal denaturation at higher temperatures.  We therefore chose to calculate Ea from Arrhenius plots done over the range in which rate increased with increasing temp. We did attempt to fit our data to the Macromolecular Rate Theory (MMRT) model descried by Alster et al., (2016), which better accounts for observed declines in enzyme activity at temperatures below denaturation temperatures and enables the derivation of an optimum temperature. However, our data did not fit this model very well.

Alster, C.J., Baas, P., Wallenstein, M.D., Johnson, N.G. and Von Fischer, J.C., 2016. Temperature sensitivity as a microbial trait using parameters from macromolecular rate theory. Frontiers in microbiology, 7, p.1821.

**I have read the manuscript titled "Differential Temperature Sensitivity of Intracellular and Extracellular Soil Enzyme Activities" by Adekanmbi et al.**

**The study has two main objectives. The first objective is to evaluate the thermal sensitivity of the extra- and intracellular steps of soil organic matter decomposition. The second objective is to evaluate the potential of microbial communities to acclimatize/adapt to a temperature treatment over 60 days.**

**The study is well written and the introduction and discussion sections are well-supported with relevant hypotheses and current literature. I find that the topic and questions raised in this article are of great interest, as there is still a lack of understanding about the thermal sensitivity of soil microorganisms, their potential to adapt to climate change, and the implications on soil carbon decomposition. The study is well-written and has a clear introduction and discussion with well-stated hypotheses and up-to-date bibliography. However, the study has three main limitations that I highlight below.**

We also thank Reviewer 2 for their positive appraisal of our work and also respond below to each of the three limitations and outline how we are happy to make revisions to address the minor comments also.

**1- I understand the idea of removing substrate limitation by feeding microbes with glucose, but using this as a proxy for intracellular enzyme activity is confusing. Other factors, such as diffusion, active transport, and carbon use efficiency of the microbes, among others, can also impact this step. Additionally, comparing the intracellular decomposition process (which involves multiple enzymes) to an extracellular specific enzyme reaction (such as beta glucosidase or chitinase) seems not appropriate. The authors should rephrase this in their manuscript and consider discussing non-limited respiration or maybe glucose-induced respiration.**

We agree with this comment (also made by Reviewer 1) that referring to all intracellular metabolic processes as 'intracellular enzyme activity' is overly simplistic. We are therefore happy to revise the manuscript (including the title) so that we refer to 'intracellular metabolic processes' when speaking generally and to 'glucose-induced respiration' when speaking specifically about our results.

**2- I do not understand why the authors are calculating Q10 at different temperatures. It is known that one of the main limitations of Q10 is that it can change depending on the temperature range chosen for calculation. Why did the authors not use linear regression between the natural logarithm of enzyme activity (Vmax) and temperature, and convert to Q10 values based on the relationship: Q10 = exp (10 × slope) (as cited in Zuo et al, 2021, German et al, 2016 and many other articles)? Can you please provide a strong rationale for why this method was not used or present a single Q10 value calculated in this manner.**

**Refs**

**-The effect of soil depth on temperature sensitivity of extracellular enzyme activity decreased with elevation: Evidence from mountain grassland belts. 2021. Yiping Zuo, Hongjin Zhang, Jianping Li, Xiaodong Yao, Xinyue Chen, Hui Zeng, Wei Wang,**

**-The Michaelis-Menten kinetics of soil extracellular enzymes in response to temperature: A cross-latitudinal study.2016.  German, D.P., Marcelo, K.R.B., Stone, M.M., Allison, S.D.**

The primary reason why we calculated Q10 at different temperatures and not using the method identified here is because we found that temperature sensitivity was different at different temperature ranges. This meant that there was not a good linear relationship between the natural log of enzyme activity/respiration and temperature apart for β-glucosidase, as demonstrated in the Figure below. We therefore felt that presenting a single Q10 representing temperature sensitivity across the range of temperatures assayed would be misleading and miss some of the nuisances in our findings.

We did attempt to fit our data to the Macromolecular Rate Theory (MMRT) model descried by Alster et al., (2016), which better accounts for non-monotonic relationships between enzyme activity and temperature. However, our data did not fit this model very well either.

Alster, C.J., Baas, P., Wallenstein, M.D., Johnson, N.G. and Von Fischer, J.C., 2016. Temperature sensitivity as a microbial trait using parameters from macromolecular rate theory. Frontiers in microbiology, 7, p.1821.

[Figure]

**3- This study only uses one soil and three temperature treatments to explore the relative thermal sensitivity of extra- and intracellular steps of decomposition. I acknowledge that determining thermal sensitivity in the laboratory is a lot of work, but using only one soil and three treatments is still very limited compared to other published studies. The authors should clearly state this limitation in the abstract and main conclusion to avoid extrapolating or overstating the main findings (which are indeed interesting).**

We agree that the use of a single soil and only 3 pre-incubation temperatures is a shortcoming that limits the extent to which the results can be generalized. We propose to address this limitation in the revision.

The abstract currently states:

> *"This result implies that depolymerisation of higher molecular weight carbon is more sensitive to temperature changes at higher temperatures (e.g. higher temperatures on extremely warm days) but the respiration of the generated monomers is more sensitive to temperature changes at moderate temperatures (e.g. mean daily maximum soil*

*temperature). Therefore, since climate change predictions currently indicate that there will be a greater frequency and severity of hot summers and heatwaves, it is possible that global warming may reduce the importance of extracellular depolymerisation relative to intracellular metabolic processes as the rate limiting step of soil organic matter mineralization."*

We propose to add a sentence to the abstract to identify the limitations of our findings so that this passage of text will read:

*"This result implies that depolymerisation of higher molecular weight carbon is more sensitive to temperature changes at higher temperatures (e.g. higher temperatures on extremely warm days) but the respiration of the generated monomers is more sensitive to temperature changes at moderate temperatures (e.g. mean daily maximum soil temperature). However, studies using multiple soil types and a greater range of pre-incubation temperatures are required to generalize our results. Nevertheless, since climate change predictions currently indicate that there will be a greater frequency and severity of hot summers and heatwaves, it is possible that global warming may reduce the importance of extracellular depolymerisation relative to intracellular metabolic processes as the rate limiting step of soil organic matter mineralization."*

The conclusion previously stated:

*"Specifically, for the grassland soil under study, we have demonstrated that potential activities of extracellular depolymerase enzymes (β-glucosidase and chitinase) have greater sensitivity to increases in temperature in the range of temperatures experienced on extremely warm days (between 26 °C and 37 °C) than the potential activity of intracellular enzymes involved in catabolism of monomeric (e.g. glucose) substrates to CO2."*

We propose to revise this passage of text in the conclusion so that it now reads:

*"Specifically, for our individual grassland soil pre-incubated at just three representative temperatures, we have demonstrated that potential activities of extracellular depolymerase enzymes (β-glucosidase and chitinase) have greater sensitivity to increases in temperature in the range of temperatures experienced on extremely warm days (between 26 °C and 37 °C) than the potential activity of intracellular enzymes involved in catabolism of monomeric (e.g. glucose) substrates to CO2."*

We believe these changes would bring the abstract and conclusion into line with the tone of the discussion which includes the following sentence:

*"further experiments are required to evaluate the applicability of our finding of a greater temperature sensitivity of extracellular activities at higher (26 °C and 37 °C) temperature ranges to other soil types"*

**Line to line comments:**

**Line 100: "measurement of enzyme activity at different temperatures in the lab is not an experimental treatment in itself (compared to the 60 day of temperature treatment). This sentence is misleading. The experiment did not have "60 experimental units," but 12 (3 incubation temperatures x 4 replicates).**

We agree that the way this is written is misleading and we propose to revise the section to address this shortcoming. The methodology currently states:

> *"The experiment was a two factorial experiment involving the 3 pre-incubation temperatures (5 °C, 15 °C, and 26 °C), and 5 assay temperatures (5 °C, 15 °C, 26 °C, 37 °C and 45 °C). This design resulted in 15 treatments replicated 4 times, resulting in 60 experimental units. At the end of the 60-day pre-incubation period, soils were subsampled for determination of basal respiration and substrate induced respiration using glucose as the substrate (Section 2.3), and the potential activity of β-glucosidase (β-1,4-glucosidase) and chitinase (N-acetyl β – D – glycosaminidase) extracellular enzymes (Section 2.4)."*

We propose to revise this section so that it reads:

> *"The experimental design included 3 pre-incubation temperatures (5 °C, 15 °C, and 26 °C), replicated 4 times, resulting in 12 experimental units. At the end of the 60-day pre-incubation period, soils were subsampled for determination of basal respiration and substrate induced respiration using glucose as the substrate (Section 2.3), and the potential activity of β-glucosidase (β-1,4-glucosidase) and chitinase (N-acetyl β – D – glycosaminidase) extracellular enzymes (Section 2.4) all measured at 5 assay temperatures (5 °C, 15 °C, 26 °C, 37 °C and 45 °C)."*

**Line 97: A space is needed between the two commas.**

Here we accidentally included the same reference twice. We propose to correct this in a revision. The current version is:

> *"by Adekanmbi et al., (2020)(Adekanmbi et al., 2020)."*

The revision will be:

> *"by Adekanmbi et al., (2020)."*

**Line 128: What does MUB stand for? Was the buffer pre-incubated at different temperatures?**

MUB stands for 4-methylumbelliferone and we propose to add this to the sentence to make the methodology clearer. The buffers were pre-incubated at room temperature prior to assays.

We propose to revise this sentence of the methodology so that it reads:

> *"For each experimental replicate, 1 g of soil was weighed into a 50 ml centrifuge tube and mixed with 4ml pre-incubated 4-methylumbelliferone (MUB) buffer"*

**Line 326/327: "Accumulation of monomers" needs to be reformulated.**

We agree that this phrasing is unclear. The text currently states:

> *"Such a switch in rate limitation, if applicable generally across all extra- and intracellular reactions, would result in an accumulation of monomers and thus potential for greater losses of C from the soil profile as dissolved organic carbon, an often overlooked component of terrestrial carbon budgets (Evans et al., 2014; Cook et al., 2018)."*

We propose to revise this text so that it reads:

> *"Such a switch in rate limitation, if applicable generally across all extra- and intracellular reactions, would result in a build-up of low molecular weight substrates in the soil and thus potential for greater losses of C from the soil profile as dissolved organic carbon, an often overlooked component of terrestrial carbon budgets (Evans et al., 2014; Cook et al., 2018)."*

**Line 343: "It is tempting" is not appropriate scientific language. Please rephrase.**

The sentence currently reads:

*"It is tempting to initially suppose that the substrates that are hydrolysed by chitinase and β-glucosidase enzymes in depolymerization reactions might be more recalcitrant than glucose and other lower molecular weight substrates for intracellular respiration and, in consequence, the extracellular-catalysed reactions should have higher temperature sensitivities."*

We propose to revise this sentence in response to the comment so that it reads:

*"It might be initially supposed that the substrates that are hydrolysed by chitinase and β-glucosidase enzymes in depolymerization reactions might be more recalcitrant than glucose and other lower molecular weight substrates for intracellular respiration and, in consequence, the extracellular-catalysed reactions should have higher temperature sensitivities."*

**Line 346: Please remove the hyphen in "trimeric."**

The sentence currently reads:

*"However, it should be recognised that chitinase and β-glucosidase have relatively simple di- or tri-meric substrates in nature and are assayed using artificial and simple substrates that may not be more recalcitrant than those used in intracellular metabolism."*

We propose to revise the sentence so that it reads:

*"However, it should be recognised that chitinase and β-glucosidase have relatively simple dimeric or trimeric substrates in nature and are assayed using artificial and simple substrates that may not be more recalcitrant than those used in intracellular metabolism."*

**Line 349: Please remove the comma.**

We think that there is an errant bracket here rather than a comma. The sentence currently reads:

*"In addition, the theoretical predictions refer to chemical decomposition reactions and not necessarily those involving enzyme catalysis (Blagodatskaya et al., 2016))."*

We propose to revise it so that it reads:

*"In addition, the theoretical predictions refer to chemical decomposition reactions and not necessarily those involving enzyme catalysis (Blagodatskaya et al., 2016)."*

We thank reviewer 2 for pointing this out.

**Line 361: Double space?**

We propose to remove the double space here and from several other places between the end of one sentence and the start of another in the revision to the manuscript.

**Line 384: Changes in the thermal sensitivity of enzymes could have indicated an adaptation of the enzymes produced by the microbial community.**

We agree with this comment but don't want to give the impression that it is the enzymes themselves that are adapting. Rather it is the microbial community that produces the enzymes that is adapting. The sentence currently reads:

*"It is likely that such indirect effects of pre-incubation temperature on microbial community composition enzyme pool size masks any direct thermal acclimatation or genetic adaptation of the soil microbial community."*

We propose to revise this sentence so that it reads:

*"It is likely that such indirect effects of pre-incubation temperature on the microbial enzyme pool size masks any direct thermal acclimatation or genetic adaptation of the soil microbial community and subsequent change in the temperature sensitivity of the enzymes it produces."*

We elaborate on this point in the next paragraph.

**Line 386: Please use "Vmax" or "apparent Vmax" instead of "concentration," as you did not measure it. Please make sure to use consistent terminology throughout the manuscript.**

Throughout the manuscript we refer to potential enzyme activity. However, in these sentences we incorrectly refer to concentration. We did not determine the relationship between rate and substrate concentration in order to estimate Vmax. We therefore propose to revise this sentence and replace 'concentration' with 'potential enzyme activity' rather than Vmax.

The sentence currently reads:

> *"Compared to β-glucosidase, there was less evidence of an effect of pre-incubation temperature on the concentration of chitinase (no significance of pre-incubation as a main effect). The concentration of an enzyme in soil is a function of production versus turnover rate."*

We propose to revise it so that it reads:

> *"Compared to β-glucosidase, there was less evidence of an effect of pre-incubation temperature on the potential enzyme activity of chitinase (no significance of pre-incubation as a main effect). The potential activity of an enzyme in soil is a function of production versus turnover rate."*

**In the supplementary material: Could you please specify if the curves on the graph are the mean of all samples or just one sample for illustrating the reaction? (Figure S-3: β-glucosidase).**

Each of the points on the graphs in the Supporting Information represent the mean of three replicates. We propose to revise the Supporting Information to explicitly state this the captions.

**Line 391: You could calculate enzyme-specific activity (normalized by microbial biomass) to test if this statement is correct or not?**

We normalized all our enzyme assay and respiration measurements to microbial biomass (shown in the figure below) and we can see here that the reason for a lower rate of intracellular metabolic processes in soils incubated at 26 °C is not due to a lower microbial biomass.

[Figure]

We therefore propose to revise the passage of text where this is mentioned as a possibility. The sentence currently reads:

> *"Whilst pre-incubation at 26 °C reduced the rate of intracellular metabolic processes, probably linked to reductions in biomass and relative C availability, it did not lead to an alteration of community intracellular temperature response traits…"*

We propose to revise it so that it reads:

> *"Whilst pre-incubation at 26 °C reduced the rate of intracellular metabolic processes, it did not lead to an alteration of community intracellular temperature response traits"*

**Figure 2 caption: Please specify that it is Vmax.**

We did not determine the relationship between rate and substrate concentration in order to estimate Vmax. Although we acknowledge this as a limitation, undertaking this task would have increased the number of assays by 5 or 6 fold. Instead we showed that there was a linear relationship between assay product formation and time in our assays (see supplementary information).  This indicates that the substrate was supplied at an initial concentration that was sufficiently in excess such that the depletion of substrate

concentration through enzymatic conversion over the assay period did not limit the reaction rate.

---

## Author Response (AR2)

**Reviewer 1**

**I have only one major point to raise:**

**In line 170 authors wrote "Ea was calculated for two ranges of temperature (5 °C – 26 °C for intracellular enzymes and basal respiration, and 5 °C – 37 °C for the two extracellular enzymes) to ensure that the data used to calculate Ea conformed to the Arrhenius functional form (Schulte, 2015)". Looking at the figures it is not clear why authors decided to use different temperatures to calculate the activation energy of intracellular and extracellular processes. I would suggest to use the same range of temperatures or justify very well why they think the ranges of temperatures should be different. As the authors want to compare the activation energy from these different processes, using distinct temperature ranges creates a problem for their comparison. Please justify your choice or make the required changes in the analysis and corresponding figure.**

Our reason for calculating Ea at 5 °C – 26 °C for intracellular metabolic processes and basal respiration, and 5 °C – 37 °C for the two extracellular enzymes was to ensure that the Ea was quantified in the rising phase of the temperature response for each assay (consistent with Arrhenius theory). However, we understand that this can be problematic when making a direct comparison between the intracellular and extracellular processes. We have therefore calculated Ea at 5 °C – 26 °C for both intracellular and extracellular processes in the revised manuscript and presented this in a revised Figure 4. The main difference is lower Ea for chitinase in soils pre-incubated at 5 °C and 15 °C. The p value for the effect of pre-incubation temperature on Ea increases from P = 0.001 to P = 0.002 but the p value for the interaction between pre-incubation temperature and enzyme type decreases from P = 0.046 to P = 0.029. β-glucosidase activity and chitinase activity had a significantly lower Ea than intracellular metabolic activity, and basal respiration.

[Figure]

**Reviewer 2**

I have gone through the revised manuscript titled "Differential Temperature Sensitivity of Intracellular and Extracellular Soil Enzyme Activities" by Adekanmbi et al. The manuscript has shown improvement, and the authors have addressed all of my comments. However, there are still some points that require correction or clarification.

1/I am surprised that the article titled "The Inflection Point Hypothesis: The Relationship between the Temperature Dependence of Enzyme-Catalyzed Reaction Rates and Microbial Growth Rates" by Erica J. Prentice et al. has not been cited in the present study, as its present theory that explain the results obtain in the present study. Therefore, I suggest that the authors cite and integrate this article into their discussion.

We thank reviewer 2 for highlighting this article to us. We have integrated this into our discussion. However, if we understand correctly, we think that it does not explain our results. The theory indicates that the optimum growth rates of organisms occur at the inflection point of the rising phase of the thermal response of intracellular enzymes. However, we don't see any consistent relationship between pre-incubation temperature and the inflection point and we observe that the inflection point is consistently at a higher temperature for extracellular enzymes than intracellular processes. Of course, this interpretation relies on the assumption that it is the same organisms producing the extracellular enzymes who also carry out the intracellular processes.

We expanded the following sentence in the previous version:

> "Our finding that intracellular catabolism increased with increasing assay temperature up to an optimal temperature between 26 °C and 37 °C, followed by a significant decline thereafter, is very likely due to the inability of the microbial population to function optimally above 37 °C due to impairments in their physiological processes (Todd-Brown et al., 2012; Maire et al., 2013)"

So that is now reads:

> "Our finding that intracellular catabolism increased with increasing assay temperature up to an optimal temperature between 26 °C and 37 °C, followed by a significant decline thereafter, is very likely due to the inability of the microbial population to function optimally above 37 °C due to impairments in their physiological processes (Todd-Brown et al., 2012; Maire et al., 2013) and uncoupling of relative rates of constituent enzymes leading to regulatory compromise (Prentice et al. 2020)."

We have also added the following passage of text to our discussion:

> "Prentice et al., (2020) identify a relationship between the inflection point of the rising phase of the thermal response of intracellular enzyme activities and the growth rate of the organism. This inflection seems to occur between 15 °C and 26 °C for intracellular metabolic processes and between 26 °C and 37 °C for extracellular enzymes in our experiment and pre-incubation temperature does not consistently affect the temperature at which this inflection point occurs (Figure 2)."

**Please also correct the title based on reviewer 1 and my comment about "intracellular enzyme" vs "intracellular metabolic processes'".**

We have corrected the title to "*Differential Temperature Sensitivity of Intracellular Metabolic Processes and Extracellular Soil Enzyme Activities*"

**2/I appreciate the authors' clarification on the reason for calculating Q10 at different temperatures and presenting graphs of the relationship between the natural log of enzyme activity/respiration and temperature. The presented graphs should be included in the article, either in the main text or as supplementary information.**

We have added these figures to the supplementary material and included the following sentence in our method:

> "*The primary reason why we calculated Q10 at different temperatures is because we found that temperature sensitivity was different at different temperature ranges. This meant that there was not a good linear relationship between the natural log of enzyme intracellular metabolic processes or extracellular enzyme activity and temperature, apart for β-glucosidase, as demonstrated in the supplementary material.*"

**It seems that beta glucosidase follows the Arrhenius function, and respiration thermal sensitivity presents a bell-shaped response similar to the one proposed by the MMRT. However, I am more concerned about the thermal response of Chintinase, which appears to be more chaotic. Therefore, I suggest that the authors add a couple of sentences that precisely describe the observed temperature relationship of Chintinase and provide arguments or hypotheses to explain why the Chintinase temperature does not follow what is generally observed.**

In the previous version of the manuscript we already provided quite a long discussion of the chitinase thermal response as follows:

> "*The potential activity of chitinase also increased with temperature, but, in contrast to β-glucosidase, the response, over the range of assay temperatures, was non-monotonic, reaching a maximum activity between 37 °C and 45 °C.  This observed non-monotonic response to increasing temperature is interpreted in terms of three distinct phases: (i) a rising phase where temperature increases lead to increasing reaction rate due to thermodynamic effects, (ii) a plateau which represents the optimum temperature and (iii) a steep falling phase where rate declines beyond the optimum temperature (Schulte, 2015) attributed to thermal denaturation of proteins.  Our optimum for chitinase (37 to 45 °C) is relatively consistent with the report of a maximum activity for soil chitinase of 45.5 °C (as assayed through quantification of N-acetyl glucosamine released from added chitin; (Rodriguez-Kabana,  et al., 1983)) but contrasts to the optimum of ~63 °C reported in the study by (Parham and Deng, 2000) using the same p-nitrophenol-based assay as used here. Differences in these optimum temperature-activity responses between soils may be due to differences in microbial composition (and thus microbial-produced chitinase isozymes) between soils. Optimum temperatures varying between  40 °C and 60 °C have been recorded for chitinases (partially) purified from soil microorganisms (Gao et al., 2008; Alster et al., 2016; Du et al., 2021; Thakur et al., 2021). Additionally, soil-type dependent stabilization of enzyme structure against thermal denaturation through interaction with soil surfaces might also mediate differential temperature responses (Sarkar et al., 1989).  It is presumed that β-glucosidase activity in our study soil had a temperature optimum beyond the maximum tested and our finding that the optimum temperature for chitinase activity was lower than that of β-glucosidase is likely due to between-enzyme family differences in protein structural*

*properties conferring thermal stability, resulting in differential susceptibility of different enzyme families to thermal denaturation or degree of stabilization in soil."*

Instead of adding more to this part of the discussion, we have added a sentence to the end of the paragraph where we introduce the inflection point hypotheses and note the following:

*"However, the chitinase enzyme activity does not clearly demonstrate the bell-shaped thermal response expected by macromolecular rate theory. This response may have been observed if assays were undertaken at more temperatures between 26 °C and 45 °C."*

**3/Was the buffer pre-incubated at different temperatures at the desired temperature? How long did it take for the assay solution to reach the desired temperature (5C or 40C)? As the activity was only assessed based on a 30 min assay, I am wondering if the measure really reflects the activity at 5 degrees or rather an average temperature between room temperature and 5 degrees during the 30 min of cooling. In the latter case, that would mean that extreme temperatures (temperatures that are most different from room temperature, 5 or 40C) are most likely wrong or closer to room temperature than expected (8 C, 35 C, for example). Moreover, did the authors control the temperature of the buffer assay solution during the 30 min, or at least control the temperature at the end?**

**Overall, the material and method section needs to be much more precise on this point, as the entire article is about the temperature sensitivity of enzymes, and the observed response of one enzyme (chitinase) does not follow proposed/common theory (Arhenius, MMRT).**

The buffer was pre-incubated at room temperature rather than at assay temperature. We have made the text more precise by stating "*1 g of soil was weighed into a 50 ml centrifuge tube and mixed with 4ml of room temperature pre-incubated 4-methylumbelliferone (MUB) buffer (pH 6)*"

Unfortunately, we cannot change this and accept that it is likely that during the first few minutes of the assay the temperature of the soil will change from room temperature to incubation temperature. We have added the following sentence to the methodology to acknowledge this shortcoming.

*"It is likely that during the first few minutes of the assay the soils were changing temperature from room temperature to the assay temperature."*

The temperate during the assay was controlled. The incubators where the incubations took place were set to the desired temperature long before the start of the assay.

**4/In the materials and methods section, the authors indicate that the concentration of the substrate used was "1ml 25mM nitrophenyl-β-D-glucopyranoside or 10 mM p-nitrophenyl-N-acetyl-b-D-glucosaminide solution." I suggest that the authors explain why they chose this concentration and whether it was chosen to be most likely in a non-substrate limiting condition (although not tested for the soil under study). In the latter case, this would mean that the authors measured Apparent Vmax.**

The protocols are based on published methods. Eivazi and Tabatabai (1988) used 25mM PNG and Parham and Deng (2000) used 10mM NAG.

We have good evidence to suggest that the concentrations chose are not substrate limiting since we carried out a preliminary experiment incubating the soil and the substrates for 15, 30, 45, and 60

minutes and observed a linear response (Figures S-3 and S-4 of the Supporting Information). Had the assay been substrate limiting, we would not have observed a linear response. The time selected for the experimental assay (30 minutes) was in the middle of this linear range.

Eivazi, F and Tabatabai, M A (1988) Glucosidases and galactosidases in soils.  Soil Biology & Biochemistry 20, 601-606

J.A. Parham, S.P. Deng (2000) Detection, quantification and characterization of b-glucosaminidase activity in soil. Soil Biology & Biochemistry 32, 1183-1190.